# Are population trends in high-risk alcohol consumption in smokers associated with trends in quit attempts and quit success? A time-series analysis

Emma Beard ,[1] Jamie Brown,[1,2] Robert West,[1] Susan Michie[3]

¹Research Department of Behavioural Science and Health, University College London, London, UK
²SPECTRUM Collaboration, London, United Kingdom
³Clinical, Educational and Health Psychology, University College London, London, UK

**Correspondence to**
Dr Emma Beard;
e.beard@ucl.ac.uk

## ABSTRACT

**Objectives** Monthly changes in the prevalence of high-risk drinking and smoking in England appear to be positively correlated. This study aimed to assess how far monthly changes in high-risk drinking were specifically associated with attempts to stop smoking and the success of quit attempts.

**Design** Data were used from the Alcohol and Smoking Toolkit Studies between April 2014 and June 2018. These involve monthly household face-to-face surveys of representative samples of ~1800 adults.

**Setting** England.

**Participants** Data were aggregated on 17 560 past-year smokers over the study period.

**Primary and secondary outcome measures** Autoregressive integrated moving average with exogenous input (ARIMAX) modelling was used to assess the association over time between monthly prevalence of high-risk drinking among smokers and (a) prevalence of attempts to quit smoking and (b) prevalence of successful quit attempts in those attempting to quit. Bayes factors (BF) were calculated to compare the null hypothesis with the hypothesis of an effect sufficiently large (β=0.6) to explain the established association between overall prevalence in smoking and high-risk drinking.

**Results** No statistically significant associations were found between monthly changes in prevalence of high-risk drinking among smokers and attempts to quit smoking (β=0.156, 95% CI −0.079 to 0.391, p=0.194) or quit success (β=0.066, 95% CI −0.524 to 0.655, p=0.827). BF indicated that the data were insensitive but suggested there is weak evidence for the null hypothesis in the case of both quit attempts (BF=0.80) and quit success (BF=0.53).

**Conclusions** Monthly changes in prevalence of high-risk alcohol consumption in England are not clearly associated with changes in quit attempt or quit success rates.

## BACKGROUND

In England, around 15% of the population are smokers and 20% drink alcohol at high-risk levels, that is, levels which are likely to cause harm.[1,2] Both are associated with a number of preventable conditions and appear to have an accumulative effect on the risk of mortality.[3] The association between

**Strengths and limitations of this study**

► This is the first time-series study to assess how far monthly changes in high-risk drinking are associated with attempts to stop smoking and the success of quit attempts.
► This study uses a large representative sample of the population in England.
► In countries with weaker tobacco control, different effects may be observed.
► Data are observational and so strong conclusions regarding cause and effect cannot be made.

high-risk drinking and smoking has been well established at an individual level. High-risk drinkers are substantially more likely to smoke[4–8] and smokers who report starting a quit attempt also report lower alcohol consumption.[9,10] Attempts to quit smoking are also less successful among those with an alcohol use disorder.[11–13] Such associations may arise by a number of mechanisms. For example, smokers drinking at high-risk levels may follow advice that it is important to restrict alcohol consumption when they quit,[9,14–16] alcohol and smoking appear to provide cues to lapses for the other and there may be pharmacological interactions between nicotine and alcohol.[17–19] This is contrary to the popular notion of self-medication and reward seeking with people deprived of cigarettes compensating by increasing their use of alcohol.[20]

It is important to identify whether similar patterns exist at a population level. An association in either direction could mean that policies that reduce smoking prevalence may have the added benefit of reducing high-risk drinking or vice versa. In England, since 2014, monthly data have been gathered on high-risk drinking, smoking status, attempts to quit smoking and quit success.[21] Recently, we used these data to examine population-level

associations over time between smoking and high-risk drinking and showed that monthly changes in prevalence of smoking in England were associated positively with prevalence of high-risk drinking. However, there were no significant associations between motivation to stop and motivation to reduce alcohol consumption or attempts to quit smoking and attempts to reduce alcohol consumption.[22] We found the combination of results surprising and suggested that the association with overall prevalence may be related to an unmeasured variable that accounted for the change in both smoking and high-risk drinking. Alternatively, the failure to find an overall association between motivation and attempts for each behaviour may be an issue of power when focussing on the global association between subsamples that represented only a fifth of the overall sample.

This study attempted to resolve this apparent contradiction and explore the previously identified positive association between prevalence of smoking and prevalence of high-risk drinking. We relied on the assessment of trends between more specific outcomes expected to be more strongly related, if the identified association between the changes in the overall prevalence of smoking and high-drinking was causal. Specifically, we will assess whether changes in trends of excessive alcohol consumption among smokers are associated with trends in attempts to quit smoking and quit success. If no association is found, this would support the conclusion of a third unmeasured variable associated with both smoking and high-risk drinking.

This study addressed the following research questions:
1. Is there an association in England between increases or decreases in monthly prevalence of high-risk drinking among smokers and attempts to quit smoking?
2. Is there an association in England between increases or decreases in monthly prevalence of high-risk drinking among smokers and quit success rates?

## METHODS
### Study design
Data were used from the Smoking and Alcohol Toolkit Studies (STS and ATS) collected between April 2014 and June 2018. The STS and ATS are ongoing studies that involve a series of monthly cross-sectional household, face-to-face, computer-assisted surveys of representative samples of ~1800 adults in England aged above 16. Thus, the same participants take part in both surveys. The respondents are recruited using a type of random location sampling, which is a hybrid between random probability and simple quota sampling. England is first split into over 170 000 'output areas', comprising of approximately 300 households. These areas are then stratified according to A Classification Of Residential Neighbourhoods (ACORN) characteristics and geographic region (http://www.caci.co.uk/acorn/) and are randomly allocated to interviewers. Interviewers visit households within their allocated locality starting at a random point in the

area. One member per household, chosen based on who the interviewer judge would best fulfil their quota requirements, is interviewed until interviewers achieve local quotas designed to minimise differences in the probability of participation. Participants appear to be representative of the population in England, having similar sociodemographic composition and smoking characteristics to large national surveys based on probability samples such as the Health Survey for England,[23] while drinking characteristics also appear similar at a regional level to other national surveys.[24] For further details, see: www.smokinginengland.info and www.alcoholinengland.info and the published protocols.[21 23]

### Participants
Data were collected on 88 122 participants over the study period. Of these, 19.9% (95% CI 19.7 to 20.2 n=17 560) reported that they had smoked in the past year. Forty-seven per cent of past-year smokers (n=8097) were men, 18.9% (n=3272) were aged 16–24, 19.7% (n=3416) were aged 25–34, 16.2% (n=2804) were aged 35–44, 17.0% (n=2946) were aged 45–54, 14.6% (n=2521) were aged 55–64 and 13.7% (n=2371) were aged above 65. Finally, 59.4% (n=10 286) were in manual occupations. Data from these participants were aggregated monthly and this forms the basis of the sample in this paper.

### Measures
#### Input series
Participants were asked whether they smoked or had smoked cigarettes (including hand rolled) daily or non-daily in the past year and to complete the Alcohol Use Disorders Identification Test (AUDIT).[25] The AUDIT identifies people who could be classified as dependent, harmful or hazardous drinkers and has demonstrated validity, high internal consistency and good test–retest reliability across gender, age and cultures.[26–29] Those scoring 8 or more were classed as high-risk drinkers. This is a common threshold for high-risk drinkers.[27 30–32] The prevalence of high-risk alcohol consumption among smokers in each month was obtained by counting the number smokers reporting an AUDIT score greater than or equal to 8.

#### Output series
Past-year smokers were then asked:
1. 'How many serious attempts to stop smoking have you made in the last 12 months? By serious attempt I mean you decided that you would try to make sure you never smoked again. Please include any attempt that you are currently making and please include any successful attempt made within the last year'.
2. 'How long did your most recent serious quit attempt last before you went back to smoking?'

The monthly prevalence of quit attempts was calculated as the number of respondents who reported having made one or more quit attempts in the past 12 months divided by the number of past-year smokers. The success rate in

each quarter was calculated as the number of respondents reporting that they were still not smoking divided by the number reporting having made a quit attempt.

## Covariates

Past-year smokers' socioeconomic status was assessed by social grade measured using the British National Readership Survey Social Grade Classification Tool[27]: AB (higher managerial, administrative or professional), C1 (supervisory or clerical and junior managerial, administrative or professional), C2 (skilled manual workers), D (semi-skilled and unskilled manual workers) and E (casual or lowest grade workers, pensioners, and others who depend on the welfare state for their income). The prevalence of smokers in lower social grades in each quarter was calculated as the proportion of past-year smokers who reported being in C2, D and E. Past-year smokers were also asked their age, with a mean estimated each month.

## Analysis

The analysis plan, data and syntax were preregistered on the Open Science Framework (https://osf.io/384gx/). An amendment was made to the analysis plan following reviewer comments to also adjust for sociodemographic variables. Variables can only be included in ARIMAX models at the aggregated level and must vary sufficiently over the study period.[33] There was insufficient variation in gender and ethnicity over the period but there was sufficient variation in mean age and the proportion of those in lower social grades, which were included. Studies have shown an increase in the age of smokers over time[34] and socioeconomic status is a strong predictor of quitting activity.[35 36]

Cases with missing data on either smoking or drinking variables were classified as missing in calculating the prevalence figures: smoking status (n=55; %=0.1), high-risk drinking status among smokers (n=202; %=1.2) and quit attempts among smokers (n=562; %=3.2). All data were analysed in R studio.

Data were weighted (see Fidler et al[23] for further details) to match the population in England and analysed using autoregressive integrated moving average with exogenous input (ARIMAX) modelling to assess the association between prevalence of high-risk drinking among smokers and (1) prevalence of attempts to quit smoking and (2) prevalence of successful attempts to quit smoking among those having made a quit attempt. ARIMAX is an extension of ARIMA analysis, which produces forecasts based on prior values in the time series (autoregressive terms; AR) and the errors made by previous predictions (moving average terms; MA). We followed a standard ARIMAX modelling approach.[37]

The ARIMAX assumption of weak exogeneity was met: past prevalence of quit attempts (p=0.747) and quit success (p=0.999) did not statistically predict the future prevalence of high-risk drinking among smokers. No outliers were identified in any of the series using an approach based on that described by Chen and Liu.[38 39]

To stabilise the variance, the series was log transformed. The augmented Dickey-Fuller test and visual inspection of the plots indicated that first-order differencing was required for both time series. First-order differencing involves calculating the change between one observation and the next. No additional seasonal differencing was required.[40]

The autocorrelation and partial autocorrelation functions were examined to determine the non-seasonal MA and AR terms. These suggested an ARIMAX(0,1,1) model for the time series predicting both prevalence of quit attempts and prevalence of quit success. This was confirmed by comparing models with different specifications using the Akaike information criterion (AIC). To identify the most appropriate transfer function for the continuous explanatory variables, the sample cross-correlation function was checked and models with varying distributed lags compared using the AIC. This suggested a lag of 0 when predicting the prevalence of quit attempts and predicting the prevalence of quit success, thus only current values and not lagged (past period) values of the input series were used to predict current values of the output series. In our previous study, prevalence of smoking was found to be associated with high-risk drinking with a distributed lag of 2.[22] Thus, additional sensitivity analyses were run with the output series lagged by an order of 2, that is, the time base was shifted back by 2 months.

The Ljung-Box test for white noise showed that the residuals for both fitted models were free of serial correlation. A number of additional model checks were also made. First, the autocorrelation terms included in the model were checked for their statistical significance. Second, it was determined whether the model residuals were normally distributed, random and independent. Finally, that the inclusion of the MA term conformed to the bounds of invertibility, that is, its value was <1.[37 38]

Bayes factors (BFs) were derived for non-significant findings using an online calculator to disentangle whether there is evidence for the null hypothesis of no effect (BF <1/3rd) or the data are insensitive (BF between 1/3rd and 3).[41 42] A half-normal distribution was assumed with a percentage change in the outcomes of interest for every percentage increase in the input series of 0.6%. This is on the basis of a previous study showing that smokers who had made a quit attempt were around 40% less likely to report that they were high-risk drinkers.[9] Strengthening the Reporting of Observational Studies in Epidemiology (STROBE) guidelines for the reporting of observational studies were followed throughout.[43]

## PATIENT INVOLVEMENT

Neither patients were involved in setting the research question or the outcome measures nor were they involved in developing plans for recruitment, design or implementation of the study. No patients were asked to advise on interpretation or writing up of results. There are no plans

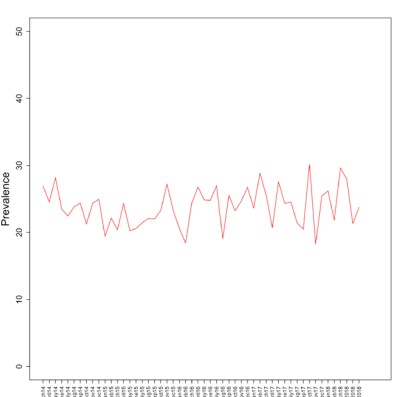
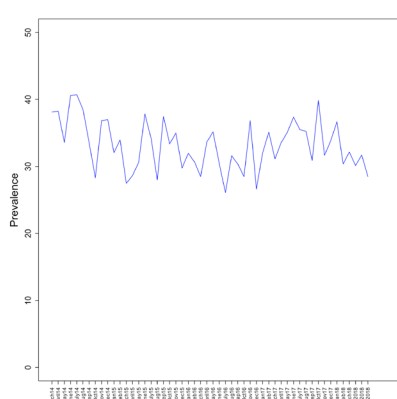
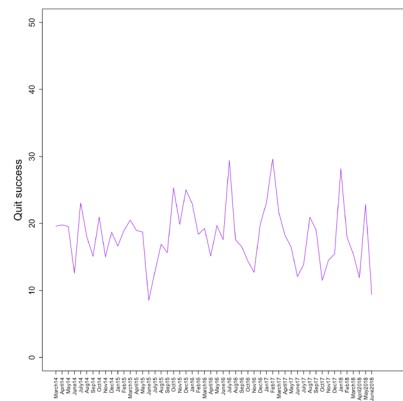

**Figure 1** Prevalence of (A) high-risk drinking, (B) attempts to quit smoking and (C) quit success.

to disseminate the results of the research directly to study participants or any specific patient community.

## RESULTS

Figure 1 shows the raw time-series data from 2014 to 2018. Prevalence of high-risk drinking among smokers declined from 26.9% (95% CI 22.3 to 32.0) in 2014 to 23.7% (95% CI 19.3 to 28.9) in June 2018. Attempts to quit smoking also declined from 38.1% (95% CI 32.7 to 43.7) to 28.5% (95% CI 23.6 to 33.9) and quit success from 19.6% (95% CI 13.2 to 27.9) to 9.4% (95% CI 4.5 to 18.0) in June 2018.

Table 1 shows the results of the ARIMAX models assessing the association between prevalence of high-risk drinking among smokers and (1) quit attempts and (2) quit success. The findings were inconclusive as to whether any associations were present. BFs suggested that there is anecdotal evidence for the null hypothesis that prevalence of high-risk drinking among smokers is not associated with prevalence of quit attempts and quit success. Findings were similar

when a 2-month back shifted lag was used for prevalence of quit attempts and quit success. Adjusting for age and social grade did not change the findings (table 2).

## DISCUSSION

To the best of our knowledge, this is the first empirical study to estimate the population association between high-risk drinking among smokers and attempts to quit smoking and the success of those attempts. There was weak evidence that there was no substantial association between changes in the prevalence of high-risk drinking and quit attempts and quit success.

These findings appear to be at odds with individual-level studies, which suggest that smokers with an alcohol use disorder are less likely to attempt and succeed in stopping smoking.[12 13] Alcohol consumption during attempts at smoking cessation is also associated with a greater risk of relapse.[14] As a result, smokers are often advised to lower their alcohol consumption when they attempt to quit smoking.[9] Of course, it remains plausible that

**Table 1** Estimated percentage point changes in proportion of quit attempts and proportion of quitters who met criteria for quit success during the study period, based on the autoregressive integrated moving average with exogenous input (ARIMAX) models

| | | Output series | | | | | |
| | | Quit attempts | | | Quit success | | |
| | | Percentage change per 1% change in the exposure | 95% CI | P value | Percentage change per 1% change in the exposure | 95% CI | P value |
|---|---|---|---|---|---|---|---|
| **Input series** | Model 1: High-risk drinking among smokers (no backward lag of the output series) | 0.156 | −0.079 to 0.391 | 0.194 | 0.066 | −0.524 to 0.655 | 0.827 |
| | Model 2: High-risk drinking among smokers (2-month backward lag of the output series) | 0.065 | −0.183 to 0.313 | 0.608 | 0.134 | −0.469 to 0.736 | 0.663 |
| | Bayes factor | | | | | | |
| | Model 1 | 0.80 | | | 0.53 | | |
| | Model 2 | 0.33 | | | 0.64 | | |

**Table 2** Estimated percentage point changes in proportion of quit attempts and proportion of quitters who met criteria for quit success during the study period, based on autoregressive integrated moving average with exogenous input (ARIMAX) models—adjusted age and socialeconomic status

| | | Output series | | | | | |
|---|---|---|---|---|---|---|---|
| | | Quit attempts | | | Quit success | | |
| | | Percentage change per 1% change in the exposure | 95% CI | P value | Percentage change per 1% change in the exposure | 95% CI | P value |
| Input series | Model 1: High-risk drinking among smokers (no backward lag of the output series) | 0.040 | −0.214 to 0.294 | 0.758 | 0.168 | −0.489 to 0.825 | 0.616 |
| | Model 2: High-risk drinking among smokers (2-month backward lag of the output series) | 0.030 | −0.229 to 0.289 | 0.822 | 0.132 | −0.549 to 0.814 | 0.703 |

high-risk drinking among smokers may still be associated with a small effect on mean population prevalence of quit attempts and their success, but it was not possible to detect this in the current study. An association may also be masked by factors impacting at a population level, which were not accounted for in the current study. Although we are unaware of any major population-level interventions or other events during the study period which may have affected the associations under investigation, we cannot rule out residual confounding. There may also be some statistical bias due to the loss of power and sensitivity that comes with the aggregating data. Prevalence of high-risk drinking among smokers will also be somewhat noisier than if prevalence was also assessed among non-smokers, given the smaller sample size involved in the estimation.

These findings suggest that the previously identified positive association between prevalence of smoking and prevalence of high-risk drinking is unlikely to be causal, whereby smokers attempting to quit, and those succeeding, also reduce their alcohol intake.[22] Although it remains possible that use of alcohol by smokers impacts on other key indices including longer term abstinence, the small proportion of smokers who relapse long term (ie, after a year) could not account for the size of association noted. It may instead be that overall prevalence is related to an unmeasured variable, perhaps economic factors and sociocultural events, that accounts for the change in both smoking and high-risk drinking. For example, in recent years, taxation on cigarettes and alcohol has increased linearly, driving down sales of both.[44 45] There have also been substantial fluctuations in average household income since 2013, which have been shown to independently affect smoking and alcohol consumption.[46–48] Sporting events such as the Olympics may also concurrently increase alcohol and tobacco intake as they are celebratory occasions. Mass media campaigns may also play role, simultaneously promoting attempts to quit smoking and the adoption of a healthier lifestyle by reducing alcohol intake.[49]

A strength of this study is the use of a large representative sample of the population in England. Several limitations need to be considered. First, the ATS required participants to recall their alcohol consumption and attempts to quit smoking which is likely to have been somewhat inaccurate due to recall bias and social desirability. For example, it has been found that a large proportion of unsuccessful quit attempts fail to be reported, particularly if they only last a short time or occurred long ago.[50] However, social pressure in population surveys tends to be low, and so it is generally considered acceptable to rely on self-reported data.[51] Secondly, these findings may not generalise to other countries. England has a strong tobacco-control climate. In countries with weaker tobacco control or different alcohol control policies, different effects may be observed. Thirdly, this paper did not consider the impact of changes in excessive alcohol consumption prevalence on the length of quit success, being defined as having made a quit in attempt in the last 12 months and still reporting not smoking. This will be an important area for future research as more data are accumulated to provide adequate power. Finally, although there can be no individual-level confounding in population trend data, there is a possibility of population-level confounding, such as introduction of policies that may affect quitting rates. However, we were unable to identify any such population policies occurring during the study period that may have confounded the results.

## CONCLUSION

These findings suggest that the previously identified positive association between prevalence of smoking and prevalence of high-risk drinking is unlikely to be causal, whereby smokers attempting to quit, and those succeeding, also reduce their alcohol intake. Instead, it may be that overall prevalence is related to an unmeasured third variable such as economic factors and sociocultural events.

**Contributors** EB, JB, SM and RW wrote the first draft of the manuscript and conducted the analysis. All other authors commented on this draft and contributed to the final version. All authors read and approved the final manuscript.

**Funding** The Smoking Toolkit Study is currently primarily funded by Cancer Research UK (C1417/A14135; C36048/A11654; C44576/A19501) and has previously also been funded by Pfizer, GSK and the Department of Health. The ATS is currently funded by the NIHR School for Public Health Research (SPHR) (SPHR-SWP-ALC-WP5). SPHR is a partnership between the Universities of Sheffield; Bristol; Cambridge; Exeter; UCL; The London School for Hygiene and Tropical Medicine; the LiLaC collaboration between the Universities of Liverpool and Lancaster and Fuse; The Centre for Translational Research in Public Health, a collaboration between Newcastle, Durham, Northumbria, Sunderland and Teesside Universities. JB's post is funded by CRUK (C1417/A14135). RW is funded by Cancer Research UK (C1417/A14135). EB is funded by the NIHR SPHR (SPHR-SWP-ALC-WP5) and CRUK also provides support (C1417/A14135).

**Disclaimer** The views expressed are those of the authors and not necessarily those of the NHS, NIHR or Department of Health. No funders had any involvement in the design of the study, the analysis or interpretation of the data, the writing of the report or the decision to submit the paper for publication.

**Competing interests** RW undertakes consultancy and research for and receives travel funds and hospitality from manufacturers of smoking cessation medications. RW salary is funded by Cancer Research UK. SM receives support from Cancer Research UK and the National Institute for Health Research (NIHR)'s School for Public Health Research (SPHR). EB and JB have received unrestricted research funding from Pfizer. PM's research is funded by a variety of governmental funding agencies including UKRI and NIHR.

**Patient and public involvement** Patients and/or the public were not involved in the design, or conduct, or reporting, or dissemination plans of this research.

**Patient consent for publication** Not required.

**Ethics approval** Ethics approval for the Smoking Toolkit Survey (STS) was originally granted by the UCL Ethics Committee (ID 0498/001) and approval for the ATS was granted by the same committee as an extension of the STS (ID 2808/005). In accordance with our ethical approval, all respondents were given a written information sheet about the study and provided informed verbal consent.

**Provenance and peer review** Not commissioned; externally peer reviewed.

**Data availability statement** Data are available in a public, open access repository. The analysis plan, data and syntax were preregistered on the Open Science Framework (https://osf.io/384gx/).

**ORCID iD**
Emma Beard http://orcid.org/0000-0001-8586-1261

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
