## [Reviewer comments · BMJ Open]

ARTICLE DETAILS

TITLE (PROVISIONAL)	Are population trends in high-risk alcohol consumption in smokers associated with trends in quit attempts and quit success? A time series analysis
AUTHORS	Beard, Emma; Brown, Jamie; West, Robert; Michie, Susan

VERSION 1 – REVIEW

REVIEWER	Sterling McPherson Elson S. Floyd College of Medicine, Washington State University; USA
REVIEW RETURNED	05-Dec-2019

GENERAL COMMENTS	Overall this article is well written, the introduction provides a brief but accurate description of the synergistic associations between tobacco and alcohol co-use; the objective is clearly defined; methods, measures and the statistical approach is strong as well. Indeed, attention being paid to the issue of missing data along with registering the study with the open science framework is a strength of this study as is the use of the STROBE checklist. The results are clear and the authors do a good job in the discussion section as well. Although the findings were inconclusive, studies that explore the interaction and consequences related to tobacco and alcohol co-use are particularly important due to the prevalence and health burden related to this co-use worldwide. One critical weakness in this study is the lack of use of any covariates. Perhaps I missed it, but it's unclear what the rationale for this was; it is a strange omission. The other critical item is the lack of congruity of time between the measures being associated with one another. I added some detail to these concerns below. I hope the below is helpful to the authors. 1. Abstract. It seems like an important limitation to note is that while quit attempts may not be associated with high risk drinking, amount of smoking may be related to amount of alcohol consumption. For example, it may be the case that those who are engaging in high risk drinking are simply not interested in reducing their smoking, but if you were to compare amount of drinking to amount of smoking, this would be a more sensitive test of the overall question.2. Abstract. If there were any pre-specified covariates included in the model, it may help to list those.3. Introduction. There is some evidence in the literature that this relationship between smoking and drinking is sex dependent. This is not mentioned in the introduction but could be one of the "third variables" that the authors are trying to potentially identify.4. Page 6 first paragraph (method section). It is not clear what criteria was used to select, among household members, a specific
---

	member to be interviewed. Was this conducted randomly or was some other criteria used to obtain a representative sample? This needs to be clarified. 5. Methods. There is no mention of covariates up to this point. Is there a reason? Given the nature of the data collected and question(s) being asked, this analysis would almost certainly necessitate the use of a variety of covariates in order to truly elucidate the true relationship. Please justify, clarify and modify the manuscript as necessary. 6. Results/Discussion. The fact that the prevalence of quit attempts and quit success measures covered a one-year period while the prevalence of high-risk alcohol consumption among smoking covered only a month may pose some bias that could have influenced the current findings. As it is currently structured, individuals who have successfully abstained from smoking in the last year are being treated as the same in the current model independent of the duration of abstinence. However, the impact of being abstinent from tobacco on high-risk drinking may vary depending on the length of tobacco abstinence. It is reasonable to consider that the impact that one week or 11 months of tobacco abstinence have on high-risk drinking tend to differ. The duration of tobacco abstinence is not being considered in the current model. This point should be further discussed and possibly included as a limitation.
--	--

REVIEWER	Theodore V Cooper University of Texas at El Paso
REVIEW RETURNED	07-Dec-2019

GENERAL COMMENTS	This manuscript has multiple strengths including an interesting research question and a large sample size. Addressing some limitations may bolster the manuscript.  1) The authors may wish to report the sample size for the study in the abstract rather than the total sample. 2) The authors may wish to more clearly indicate how this paper differs from the previously published one. 3) The authors may wish to include information about human subjects review and consent. 4) Of most concern, the authors do not seem to use an established measure for successful cessation. The current one is of variable length, rather than point prevalence abstinence, or some clear indicator of length of continuous abstinence. 5) Participant characteristics are not indicated within the manuscript or tables. Thus, it is unclear the sex, age, ethnocultural make up of the sample, as well as drinking and smoking levels at baseline, each time point. This makes interpreting the results and discussion more than challenging. 6) The authors posit economic and/or sociocultural events as potentially driving the non findings. However, other explanation could be noted (anti tobacco media, other policy changes than taxation). 7) minor grammatical and typographical errors exist that the authors may wish to attend to.
---

VERSION 1 – AUTHOR RESPONSE

Reviewer 1

1. Overall this article is well written, the introduction provides a brief but accurate description of the synergistic associations between tobacco and alcohol co-use; the objective is clearly defined; methods, measures and the statistical approach is strong as well. Indeed, attention being paid to the issue of missing data along with registering the study with the open science framework is a strength of this study as is the use of the STROBE checklist. The results are clear and the authors do a good job in the discussion section as well. Although the findings were inconclusive, studies that explore the interaction and consequences related to tobacco and alcohol co-use are particularly important due to the prevalence and health burden related to this co-use worldwide.

Thank you for these positive comments.

2. It seems like an important limitation to note is that while quit attempts may not be associated with high risk drinking, amount of smoking may be related to amount of alcohol consumption. For example, it may be the case that those who are engaging in high risk drinking are simply not interested in reducing their smoking, but if you were to compare amount of drinking to amount of smoking, this would be a more sensitive test of the overall question.

We agree that the association between mean alcohol consumption at a population level and mean cigarette consumption would be another interesting question. However, in this paper we were specifically interested in excessive alcohol assumption and attempts to stop. We had two reasons: i) there is literature on the association between alcohol consumption and quitting behaviour at the individual-level; ii) we have previously reported a population-level association between overall prevalence of smoking and high risk drinking, which we had intended the current study to elucidate.

3. Abstract. If there were any pre-specified covariates included in the model, it may help to list those. There were no covariates. Please see our response to question 6 below.

4. Introduction. There is some evidence in the literature that this relationship between smoking and drinking is sex dependent. This is not mentioned in the introduction but could be one of the “third variables” that the authors are trying to potentially identify.

The tested associations were estimated from variables aggregated at the population level over time. Unfortunately, there were not sufficient changes in the monthly sex profile over the 4-year period studied.

5. Page 6 first paragraph (method section). It is not clear what criteria was used to select, among household members, a specific member to be interviewed. Was this conducted randomly or was some other criteria used to obtain a representative sample? This needs to be clarified.

They ask who is in and then invite one person who they judge to best fulfill their quota requirements. We now include the following sentence in the design section: “One member per household, chosen based on who the interviewer judge would best fulfil their quota requirements, is interviewed until interviewers achieve local quotas designed to minimise differences in the probability of participation.”

6. Methods. There is no mention of covariates up to this point. Is there a reason? Given the nature of the data collected and question(s) being asked, this analysis would almost certainly necessitate the use of a variety of covariates in order to truly elucidate the true relationship. Please justify, clarify and modify the manuscript as necessary.

Population trend data, by definition, involve the whole population and there can be no individual-level

confounding. There is the possibility of population-level confounding, such as introduction of policies that may affect overall quitting rates. However, we were unable to identify any relevant policies during the study period. We now include the follow in the limitations of the discussion “Finally, although there can be no individual-level confounding in population trend data there is a possibility of population-level confounding, such as introduction of policies that may affect quitting rates. However, we were unable to identify any such population policies occurring during the study period that may have confounded the results.”

7. Results/Discussion. The fact that the prevalence of quit attempts and quit success measures covered a one-year period while the prevalence of high-risk alcohol consumption among smoking covered only a month may pose some bias that could have influenced the current findings. As it is currently structured, individuals who have successfully abstained from smoking in the last year are being treated as the same in the current model independent of the duration of abstinence. However, the impact of being abstinent from tobacco on high-risk drinking may vary depending on the length of tobacco abstinence. It is reasonable to consider that the impact that one week or 11 months of tobacco abstinence have on high-risk drinking tend to differ. The duration of tobacco abstinence is not being considered in the current model. This point should be further discussed and possibly included as a limitation.

We agree that this is an important point and have thus added the following as a limitation: “Thirdly, this paper did not consider the impact of changes in excessive alcohol consumption prevalence on the length of quit success, being defined as having made a quit in attempt in the last 12 months and still reporting not smoking. This will be an important area for future research as more data are accumulated to provide adequate power.”

Reviewer 2

Reviewer Name: Theodore V Cooper

1. The authors may wish to report the sample size for the study in the abstract rather than the total sample.

We have changed this to “Data were aggregated on 17,560 past year smokers over the study period”.

2. The authors may wish to more clearly indicate how this paper differs from the previously published one.

We state this at the end of the introduction but have changed some of the wording to make this clearer “This study attempted to resolve this apparent contradiction and explore the previously identified positive association between prevalence of smoking and prevalence of high-risk drinking. We relied on the assessment of trends between more specific outcomes expected to be more strongly related, if the identified association between the changes in the overall prevalence of smoking and high-drinking was causal. Specifically, we will assess whether changes in trends of excessive alcohol consumption among smokers are associated with trends in attempts to quit smoking and quit success.”

3. The authors may wish to include information about human subjects review and consent.

Information on consent can be found under the declarations at the end of the paper

4. Of most concern, the authors do not seem to use an established measure for successful cessation. The current one is of variable length, rather than point prevalence abstinence, or some clear indicator of length of continuous abstinence.

We have relied upon this outcome in this survey for the individual-level assessment of real-world effectiveness of smoking cessation treatments in England in a number of previous papers. We have argued previously [<https://www.ncbi.nlm.nih.gov/pmc/articles/PMC4171752/>] “A related issue is the assessment of abstinence by asking respondents whether they were ‘still not smoking’. This definition classified as abstinent those who had one or more lapses but resumed not smoking. This limitation would be serious if the rate of lapsing was associated with method of quitting, and should be assessed in future studies. By contrast, advantages of this measure were the assessment of prolonged abstinence, as advocated in the Russell Standard, and a clear relationship to the quit attempt in question. An alternative approach, with a view to survival analysis, may have been to assess the length of abstinence since quit date among all respondents, including those who had relapsed by the time of the survey. However, this assessment would have added noise and potential bias with smokers needing to recall the time of relapse and having different interpretations of their return to smoking (i.e. first lapse, daily but reduced smoking, or smoking at pre-quit level). The strength of our approach is that smokers only needed to know whether they were currently still not smoking.”

5. Participant characteristics are not indicated within the manuscript or tables. Thus, it is unclear the sex, age, ethnocultural make up of the sample, as well as drinking and smoking levels at baseline, each time point. This makes interpreting the results and discussion more than challenging.

Data were aggregated on a representative sample of the adult population in England and therefore individual level data were not used. We have however included details of the socio-demographic characteristics of participants under the ‘Participants’ section “Forty-seven percent of past year smokers (n=8097) were male, 18.9% (n=3272) were aged 16-24, 19.7% (n=3416) were aged 25 to 34, 16.2% (n=2804) were aged 35 to 44, 17.0% (n=2946) were aged 45 to 54, 14.6% (n=2521) were aged 55 to 64 and 13.7% (n=2371) were aged 65+. Finally, 59.4% (n=10286) were in manual occupations.”

6. The authors posit economic and/or sociocultural events as potentially driving the non findings. However, other explanation could be noted (anti tobacco media, other policy changes than taxation).

We have now included the following sentence “Mass media campaigns may also play role, simultaneously promoting attempts to quit smoking and the adoption of a healthier lifestyle by reducing alcohol intake (45).”

VERSION 2 – REVIEW

REVIEWER	Sterling McPherson Washington State University, WA; USA
REVIEW RETURNED	16-Jan-2020

GENERAL COMMENTS	The authors have addressed all of my main concerns or provided a reasonable explanation for not including some of my suggestions. One point I would make however, regards the lack of use of covariates in the models. I understood the author's argument for not using covariates since they are working with population trend data, and it's quite clear in the title, abstract and text that this was in fact the main objectives of this study. Having said that, I also believe that including covariates such as age, sex, race, education
--

	and socioeconomic status would improve the paper in a number of ways, including but not limited to:  1. Enable authors to truly elucidate the relationship between high-risk alcohol consumption and smoking quitting attempts and success. 2. Identify key covariate effects on this relationship, which in turn might impact the development of future public policy or future examinations of downstream health effects. I would strongly encourage the authors to rerun the same models with the inclusion of these covariates (and possibly other covariates that the authors might deem pertinent) to check if their findings hold and to explore the importance of these covariates. Depending on the results, this extra step could be briefly discussed in the results and discussion section and would add additional strength to this study.
--	---

REVIEWER	Theodore Cooper University of Texas at El Paso, US
REVIEW RETURNED	29-Jan-2020

GENERAL COMMENTS	The authors appear to have attended to reviewer concerns, thus, strengthening the manuscript.
---

VERSION 2 – AUTHOR RESPONSE

Reviewer 1

1. One point I would make however, regards the lack of use of covariates in the models. I understood the author's argument for not using covariates since they are working with population trend data, and it's quite clear in the title, abstract and text that this was in fact the main objectives of this study. Having said that, I also believe that including covariates such as age, sex, race, education and socioeconomic status would improve the paper in a number of ways, including but not limited to: 1. Enable authors to truly elucidate the relationship between high-risk alcohol consumption and smoking quitting attempts and success. 2. Identify key covariate effects on this relationship, which in turn might impact the development of future public policy or future examinations of downstream health effects. I would strongly encourage the authors to rerun the same models with the inclusion of these covariates (and possibly other covariates that the authors might deem pertinent) to check if their findings hold and to explore the importance of these covariates. Depending on the results, this extra step could be briefly discussed in the results and discussion section and would add additional strength to this study.

Although we used aggregated population level trend data which involve the whole population and are therefore limited by the possibility of population-level confounding and not individual-level confounding, we have included two additional covariates 1) mean age of past year smokers each month and 2) and the proportion of past-year smokers of lower socio-economic status. There was not enough variability in gender and ethnicity over time to include these as additional covariates in the ARIMAX model. We now include the following amendment in the analysis section "The analysis plan, data and syntax were preregistered on the Open Science Framework (<https://osf.io/384gx/>). An amendment was made to the analysis plan following reviewer comments to also adjust for socio-demographic variables. Variables can only be included in ARIMAX models at the aggregated level and must vary sufficiently over the study period (33)). There was insufficient variation in gender and ethnicity over the period but there was sufficient variation in mean age and the proportion of those in lower social-grades, which were included. Studies have shown an increase in the age of smokers

over time (34) and socio-economic status is a strong predictor of quitting activity (35, 36).” We have included the results of this additional analysis in Table 2.

VERSION 3 – REVIEW

REVIEWER	Sterling McPherson and Andre Miguel Washington State University Elson S. Floyd College of Medicine, USA
REVIEW RETURNED	03-Mar-2020
GENERAL COMMENTS	Authors have answered all my main concerns. This article is well written, with adequate methodology and relevant findings. As such, I support its publication at BMJ open.